# VideoEval: Vision-Centric Benchmark Suite for Low-Cost Evaluation of Video Foundation Model

## Abstract

With the accumulation of high-quality data and advancements in visual pretraining paradigms, recent Video Foundation Models (VFMs) have made significant progress, demonstrating remarkable performance on popular video understanding benchmarks. However, conventional benchmarks (e.g. Kinetics) and evaluation protocols are limited by their relatively poor diversity, high evaluation costs, and saturated performance metrics. In this work, we introduce a comprehensive benchmark suite to address these issues, namely **VideoEval**. We establish the **Vid**eo **T**ask **A**daption **B**enchmark (**VidTAB**) and the **Vid**eo **E**mbedding **B**enchmark (**VidEB**) from two perspectives: evaluating the task adaptability of VFMs under few-shot conditions and assessing their feature embedding's direct applicability to downstream tasks. With VideoEval, we conduct a large-scale study of 20 popular open-source vision foundation models. Our study reveals some insightful findings, 1) overall, current VFMs exhibit weak generalization across diverse tasks, 2) increasing video data, whether labeled or in video-text pairs, does not necessarily improve task performance, 3) the effectiveness of some pre-training paradigms may not be fully validated in previous benchmarks, and 4) combining different pre-training paradigms can help develop models with better generalization capabilities. We believe this study serves as a important complement to the current evaluation methods for VFMs and offers valuable insights for future research directions.

## 1 Introduction

The field of deep learning is experiencing a significant paradigm shift due to the emergence of foundation models (FMs). These models, exemplified by BERT [1], GPT [2, 3, 4], CLIP [5] and Stable Diffusion [6], are trained on massive and diverse data at scale and demonstrate remarkable adaptability to a broad spectrum of downstream tasks.

In the realm of video understanding, early researchers train backbone networks [7, 8, 9, 10] using visual classification tasks on large-scale labeled datasets like ImageNet [11] and Kinetics [12]. However, the high cost associated with labeled data promotes the development of self-supervised learning methods that capitalize on unlabeled data for visual pre-training [13, 14, 15, 16, 17]. Furthermore, researchers delve into multimodal pre-training utilizing large-scale visual-text pairs [18, 19, 20, 21], thereby enhancing their models' capabilities and demonstrating impressive zero-shot performance. Overall, fueled by the accumulation of high-quality image and video data and advancements in visual pre-training paradigms, Video Foundation Models (VFMs) witness remarkable progress in recent years. A new generation of VFMs [15, 16, 22, 23, 24, 25, 26] emerges, demonstrating outstanding performance on conventional video understanding benchmarks.

The rapid development of VFMs raises the problem: ***How to evaluate a video foundation model?*** In image realm, Previous works assess the generalization capability of Image Foundation Models

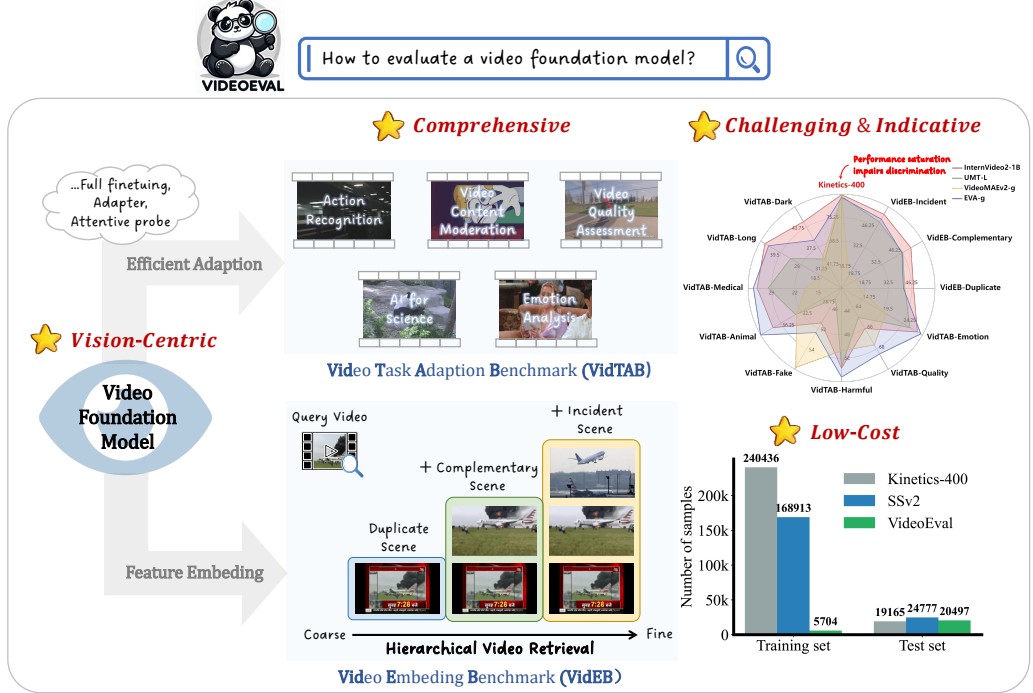

Figure 1: **Overview of VideoEval.** We propose a novel, vision-centric evaluation method for video foundation models that is comprehensive, challenging, indicative, and low-cost.

(IFMs) by evaluating their performance on numerous downstream visual tasks, encompassing diverse scenarios and evaluation protocols [27, 28, 29, 30, 31, 32, 33]. However, previous works primarily evaluates VFMs through benchmarks focusing on action recognition tasks [16, 23, 34]. Some studies [25, 26, 24] have also considered combining language models to evaluate performance on multimodal tasks. There are **several problems with current evaluation methods**: **(1)** Benchmarks like Kinetics [12], Something [35] and AVA [36], which focus on action recognition, overlook other video understanding scenarios (e.g., video quality assessment), limiting their applicability in evaluating the generalization capabilities of visual foundation backbones across diverse video understanding applications. **(2)** The performance of VFMs on conventional benchmark [37] has reached a saturation point (90% Top-1 accuracy), making it challenging to differentiate between the true capabilities of different VFMs. **(3)** The high validation costs associated with conventional evaluation protocols, which often necessitate end-to-end training on the entire dataset, pose a significant challenge, particularly for large VFMs. **(4)** Incorporating language models may introduce bias when evaluating VFMs, as performance differences might stem from the language model rather than the VFMs itself.

To tackle these problems, we build a comprehensive benchmark suite for evaluation of VFMs, namely VideoEval. As shown in Figure 1, our method has the following key features: ***Comprehensive***: First, we created the Video Task Adaptation Benchmark (VidTAB) to evaluate the adaptability of VFMs to unseen tasks with limited samples. We collected public datasets from various video task domains, including action recognition in special scenarios, AI for science, video content moderation, video quality/aesthetic assess, and emotion analysis. From these domains, we constructed eight adaptation tasks and developed evaluation protocols and adaptation methods suitable for current VFMs. Additionally, to assess the capability of VFMs' feature embedding for downstream applications, we created the Video Embedding Benchmark (VidEB), which includes four tasks that evaluate embedding at different granularities. ***Challenging & Indicative***: Due to the diversity of test data and the effectiveness of our evaluation protocols, our VideoEval can effectively distinguish between various VFMs that perform similarly on traditional benchmarks, providing deeper insights into their true capabilities. ***Low-cost***: Thanks to our training-light few-shot evaluation and training-free feature embedding evaluation protocols, VideoEval requires significantly fewer training samples compared to previous benchmarks, while maintaining a comparable number of testing samples to ensure accurate and stable evaluations. ***Vision-centric***: Our evaluation focuses solely on the Video FMs themselves, avoiding the introduction of biases that may arise from incorporating language models.

Based on VideoEval, we evaluate 20 open-source vision foundation models, including VFMs, Image Foundation Models (IFMs), and IFMs with image-to-video methods. **Our main findings as following:** First, current VFMs still struggle to adapt to unseen video tasks with limited training samples. Second, while more data and larger models generally improve performance, augmenting video training data can sometimes negatively affect certain tasks. Third, the effectiveness of certain pre-training paradigms, such as VideoMAEv2 [22], may not have been adequately validated in previous benchmarks. Finally, combining multiple pre-training paradigms can lead to models with better generalization capabilities, such as performing multimodal contrastive learning after unimodal visual self-supervised pre-training [21, 26].

Table 1: **Comparison of VFMs Benchmark.** "Num. training" denotes number of training samples, "Num. test" denotes number of test samples, and "Beyond Action" denotes the tasks in this benchmark extend beyond action understanding. Compared to previous benchmarks, our VideoEval framework achieves more comprehensive and reliable evaluations at a lower cost.

| Benchmark | Num. training | Num. test | Beyond Action | Task Diversity | Domain Diversity | VFMs-specific protocol |
|---|---|---|---|---|---|---|
| *Single-dataset Benchmarks* | | | | | | |
| Kinetics-400 [37] | 240,436 | 19,165 | ✗ | ✗ | ✗ | ✗ |
| Sth-Sth V2 [38] | 168,913 | 24,777 | ✗ | ✗ | ✗ | ✗ |
| Moment-in-Time [39] | 791,246 | 33,898 | ✗ | ✗ | ✗ | ✗ |
| UCF101 [40] | 9,537 | 3,783 | ✗ | ✗ | ✗ | ✗ |
| *Multi-dataset Benchmarks* | | | | | | |
| SEVERE [41] | 868,446 | 144,830 | ✗ | ✓ | ✓ | ✗ |
| BEAR [42] | 240,236 | 140,436 | ✗ | ✓ | ✓ | ✗ |
| VideoGLUE [34] | 1,896,621 | 239,011 | ✗ | ✓ | ✓ | ✓ |
| **VideoEval** | 5,704 | 20,497 | ✓ | ✓ | ✓ | ✓ |

## 2 Related work

**Video foundation models** With the continuous growth of image [43, 44, 45] and video data [46, 20, 47, 48, 49] and advancements in pre-training paradigms, research on Video Foundation Models (VFMs) has progressed rapidly. Current VFMs are primarily built around two pre-training paradigms: masked video modeling based on unimodal video data [15, 16, 22, 17, 50, 51, 52] and video-text contrastive learning based on multimodal visual-text pairs [18, 53, 19, 54, 20]. Some works [25, 21, 24] combine these paradigms, enabling VFMs to extend further into multimodal understanding. Additionally, some studies introduce modalities like audio and speech on top of video and text [47, 48, 26], further expanding the capabilities of VFMs. Recently, InternVideo2 [26] leverages mature pre-training paradigms and large-scale high-quality data to scale VFMs to 6 billion parameters, achieving remarkable performance improvements.

**Evaluation of VFMs** Previous works primarily utilize action recognition benchmarks focused on appearance and motion [12, 38, 36] to evaluate VFMs. To enhance evaluation diversity, some studies explore richer domains and tasks [55, 42, 56], but they remain limited to action recognition tasks. The InternVideo series [25, 26] and VideoGLUE [34] attempt to provide a more comprehensive evaluation of VFMs by expanding the number of benchmarks and evaluation protocols. However, these efforts are still based on existing benchmarks and incurred high validation costs. In contrast, our work considers the characteristics and application scenarios of VFMs, offering a comprehensive and low-cost evaluation solution through task definition and evaluation protocols, aimed at rapidly verifying the generalization capabilities of VFMs—a crucial aspect currently lacking in the community's development of these models.

## 3 Building VideoEval

We argue that a powerful video foundation model should possess two key capabilities: (1) strong task adaptation ability, i.e., the ability to ***adapt to diverse, unseen tasks with limited training samples***, and (2) the capacity to ***extract feature embedding that retain and distill key information from videos***, directly supporting various downstream tasks. From these perspectives, we construct VideoEval, which includes the Video Task Adaptation Benchmark (VidTAB) and the Video Embedding Benchmark (VidEB). By creating diverse task scenarios and employing efficient evaluation methods, VideoEval can quickly and comprehensively assess the generalization ability of VFMs in video understanding. In

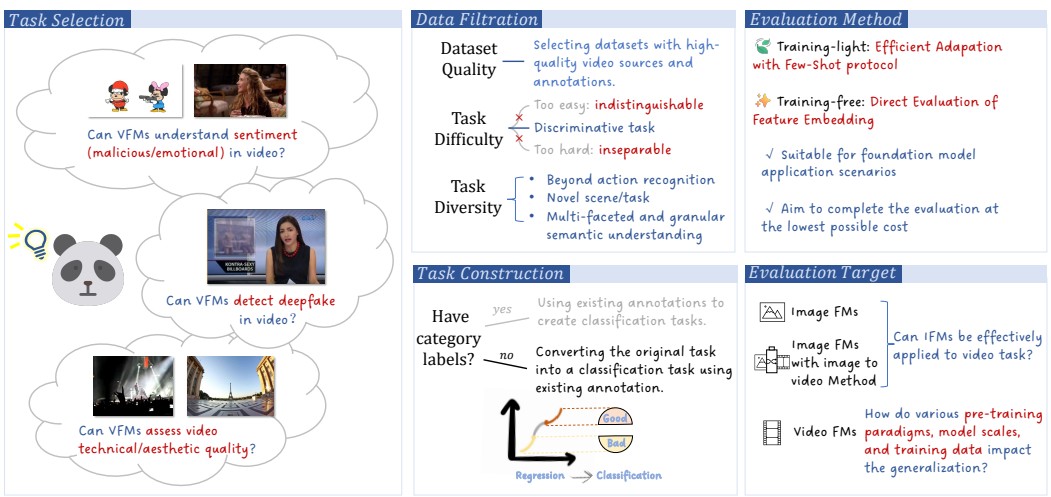

Figure 2: **Illustration of building VideoEval.**

Table 2: **Task details of VideoEval.** All videos are collected from the public datasets for building tasks of VidTAB and VidEB.

| Domain | Task | Source | Task Description |
|---|---|---|---|
| *Video Task Adaptation Benchmark (VidTAB)* | | | |
| **Action Recognition in Special-Scenarios** | Action Recognition in Dark Scene | ARID [57] | *Recognizing 11 distinct human actions in dark scenarios.* e.g. Run / Walk / Drink |
| | Action Recognition in Long Video | BreakFast [58] | *Classifying 10 types of long-duration cooking videos.* e.g. Milk / Tea / Sandwich |
| **AI for Science** | Medical Surgery | SurgicalActions160 [59] | *Classifying 16 surgical actions in gynecologic laparoscopy.* e.g. Knotting / Suction / Injection |
| | Animal Behavior | Animal Kingdom [60] | *Classifying 12 behaviors of wild animals from diverse environmental footage.* e.g. Flying / Chirping / Preening |
| **Video Content Moderation** | Fake Face | FaceForensics++ [61] | *Determine whether the faces in the video have been tampered with by AI technology (such as DeepFake).* e.g. Origin video / Video with fake face |
| | Harmful Content | mob [62] | *Detecting 3 degrees of malicious content within videos.* e.g. Obscene / Indecent activity / Violent activity |
| **Video Quality Assessment** | Quality Assess | DOVER [63] | *Evaluating videos from an aesthetic and technical perspective and categorizing them into low and high quality.* e.g. Low quality / High quality |
| **Emotion Analysis** | Emotion Analysis | CAER [64] | *Classifying 7 different human emotions in video.* e.g. Happy / Fear / Anger |
| *Video Embedding Benchmark (VidEB)* | | | |
| **|-|-|- Scene Understanding in Temporal Contexts |-|-|-** | Duplicate Scene Retrieval | FIVR5K [65] | *Retrieve Duplicate Scene Videos (DSV):* Videos captured by the same camera and sharing at least one scene (without considering any application transformations). |
| | Complementary Scene Retrieval | FIVR5K [65] | *Retrieve Complementary Scene Videos (CSV):* Retrieve a portion of the same spatiotemporal segment captured from different perspectives. |
| | Incident Scene Retrieval | FIVR5K [65] | *Retrieving Incident Scene Videos (ISV):* The same event is close in both space and time, but there are no overlapping videos. |
| | Copy Detection | DVSC23 [66] | *Detecting edited versions of the same source video.* Given a query inserted with one or more copied segments, detect the source video from the database. |

this section, we present our VideoEval in detail. The construction pipeline for VideoEval is illustrated in Figure 2, and the evaluation tasks we ultimately constructed are presented in Table 2.

### 3.1 Video Task Adaption Benchmark

**Collecting diverse dataset from public source.** Previous benchmarks primarily focus on evaluating video models based on human actions, overlooking many other tasks requiring video understanding. Therefore, we consider five different application scenarios: **1)** *Action Recognition in Special Scenarios* (**Action**): While previous benchmarks have extensively examined action recognition tasks, our focus here is to assess VFMs' capabilities in recognizing actions within special scenarios. **2)** *AI for Science* (**Science**): Referencing previous work [24], we classify tasks related to medicine and natural sciences as a category. **3)** *Video Content Moderation* (**Safety**): We group tasks related to identifying harmful or misleading information in video content. **4)** *Video Quality Assessment* (**Quality**): We categorize more subjective tasks into this group. The goal is to assess VFMs' ability to learn low-level information and human aesthetic preferences. **5)** *Emotion Analysis* (**Emotion**): We group tasks related to human emotion analysis into this category to evaluate VFMs' ability to understand human emotions.

**Constructing the adaptation task based on the existing annotations.** Classification tasks are straightforward and well-defined, with strong classification performance often indicating robust feature learning. Therefore, they are suitable for evaluating video foundation models. We construct adaptation classification tasks based on the collected data and annotations as follow: **1) Remove Low-Quality Video Datasets**: We manually exclude datasets with videos that have low resolution (below

Table 3: **Task difficulty assessment based on visual language models**. For tasks with fewer categories, such as Fake Face (n=2) and Quality Assess (n=2), random guessing can lead to high accuracy, which may result in a lower apparent proportion of hard samples. Therefore, the zero-shot classification accuracy of the models should also be considered when making task selection.

| ratio % | Dark Scene | Long Video | Medical Surgery | Animal Behavior | Fake Face | Harmfull Content | Quality Assess | Emotion Analysis |
|---|---|---|---|---|---|---|---|---|
| Easy | 18.45 | 24.57 | 0.00 | 19.18 | 39.06 | 28.78 | 53.04 | 7.21 |
| Spatial | 19.00 | 20.44 | 4.17 | 20.86 | 20.72 | 24.56 | 51.24 | 5.01 |
| Temporal | 20.09 | 22.39 | 19.79 | 23.90 | 4.89 | 22.76 | 13.26 | 27.06 |
| Hard | 36.90 | 26.28 | 62.50 | 35.58 | 9.00 | 20.17 | 3.04 | 47.15 |

240p), low frame rate (below 15fps), insufficient quantity (fewer than 150 videos per category), or low annotation accuracy (below 90%). **2) Select Discriminative Tasks**: For task difficulty screening, we first evaluate zero-shot classification performance using CLIP-L [5], EVA-g [67], ViCLIP-L [20], and Internvideo2-1B [26]. We then classify samples as follows: *Easy*: Samples that are correctly classified by three or more models. *Spatial*: Samples that are correctly classified by both CLIP and EVA. *Temporal*: Samples that are correctly classified by at least one of ViCLIP or Internvideo2-1B, but not by CLIP and EVA. *Hard*: Samples that are incorrectly classified by all models. We use the zero-shot classification accuracy of the models and the aforementioned proportions as references for task selection. Based on this, we choose tasks with lower zero-shot classification accuracy, higher proportions of Hard and Temporal samples, and lower proportions of Easy samples. The proportions of each type of sample in the tasks we ultimately selected can be found in Table 3. **3) Control the Number of Categories**: For datasets that originally include category labels, such as ARID [57] and Animal Kingdom [60], we select categories with sufficient samples to ensure evaluation accuracy and stability. We also control the final number of categories to avoid making the adaptation task overly difficult. We observed that both zero-shot testing and few-shot experiments based on current VFMs show that when the number of categories is too high, models often perform no better than random guessing. Although this issue may be mitigated as VFMs improve, we currently need to control the number of categories to effectively showcase differences between models. We select the main categories for each task and limit the number of categories to around 10 (based on few-shot experiments). **4) Handling Multi-label and Regression Tasks**: For datasets that are not originally classification tasks, we transform the tasks into classification tasks. For example, for DOVER [63], which is used for video aesthetics and technical quality assessment (a regression task), we assume that videos with quality scores in the top 40% are "high-quality videos" and those with scores in the bottom 40% are "low-quality videos", thus converting the original task into a binary classification task. In total, we construct eight classification tasks to evaluate the adaptation capabilities of video foundation models.

**Determining the evaluation protocol.** Previous studies [25, 26, 34] typically train video models using entire samples of training set, and most popular benchmarks have large training sample sizes. We argue that this evaluation method overlooks the examination of the adaptation capability of VFMs. As illustrated in Figure 3, under the scenario of using full training samples, the differences between VFMs are difficult to discern. However, under a low-sample protocol, different foundation models exhibit varying degrees of task adaptation capabilities. We observe that for tasks such as Action Recognition in Dark Scenes, which VFMs usually excel at, there are significant differences in adaptation capabilities among different models when training samples are extremely limited (4 shot and 16 shot). As the number of samples gradually increases to 100 shot, these differences diminish. Conversely, for more challenging tasks like Emotion Analysis, the performances of different models are uniformly weak when training samples are extremely limited, showing no discernible differences until a certain number of training samples (100 shot) are reached, at which point different models begin to demonstrate distinct adaptation capabilities. Therefore, to account for the adaptation capabilities of models with different numbers of training samples, we define a task adaptation capability evaluation score (TA-score):

$$TA - score = \frac{Acc^{4s} + Acc^{16s} + Acc^{100s}}{3} \tag{1}$$

Where $Acc^{4s}$, $Acc^{16s}$, $Acc^{100s}$ represent the model's top-1 accuracy for 4-shot, 16-shot, and 100-shot classifications, respectively. Unless otherwise specified, we will use TA-score to denote the performance of various tasks in VidTAB.

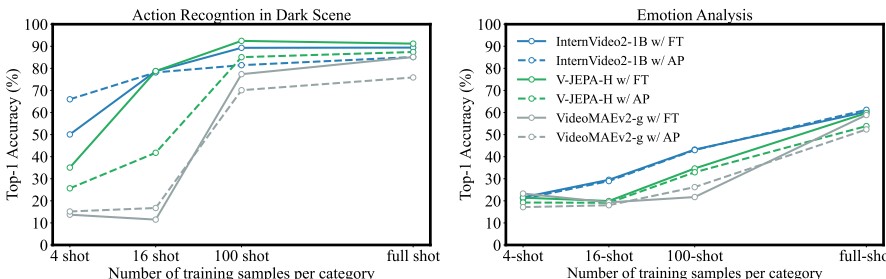

Figure 3: **Performance comparison on different training data scales.** We evaluate the performance variation of multiple video foundation models across tasks from two different domains as the scale of the training data changed. 'FT' and 'AP' denote full finetuning and attentive probe, respectively.

Table 4: **Comparison of adaptation method** All results are obtained using A100-80G with PyTorch-builtin mixed precision, using a batch size of 4 to measure Cuda memory and training time. "Dark" and "Emotion" denote the tasks of Action Recognition in Dark Scenes and Emotion Analysis, respectively. We show the result of V-JEPA-H [23] here,

| Adaptation method | Tunable Params (M) | Cuda Memory (G) | Training Time (h) | Dark TA-score | Emotion TA-score |
|---|---|---|---|---|---|
| full finetuning | 663.7 | 52.1 | 1.0 | 68.8 | 25.3 |
| adapter | 52.6 | 45.0 | 1.0 | 62.4 | 24.7 |
| attentive probe | 19.7 | 6.4 | 0.4 | 54.7 | 23.8 |
| linear probe | 0.0 | 6.0 | 0.3 | 12.9 | 16.2 |

**Identifying efficient adaptation method for evaluation.** We also need to identify how to adapt the foundation models to the corresponding task. Previous work [68, 69, 70, 71, 72] has explored various strategies for efficient adapting the foundation models. Here, we consider several of the most common and popular methods: **Full Finetuning**: Fine-tuning all the parameters of the pre-trained model. **Adapter**: Freezing the pre-trained model and inserting learnable low-rank adapter [73] modules into each block of the pre-trained model for adaptation. **Attentive Probe**: Freezing the pre-trained model and adding an additional learnable cross-attention block at the end of the model to achieve attentive pooling, followed by a linear projection for classification. **Linear Probe**: Directly using the features from the pre-trained model, performing mean pooling, and then using a linear projection for classification. We evaluate the performance of these adaptation methods based on the V-JEPA-H model, as shown in Table 4. Full finetuning and adapter exhibited the best adaptation performance, but incurred high training costs. Linear probe was highly efficient but showed weak adaptation performance. Attentive probe offered a good trade-off between efficiency and adaptation performance. Therefore, in subsequent evaluation experiments, we employed attentive probe to adapt various vision foundation models.

## 3.2 Video Embedding Benchmark

The main application domains of video embeddings we considering include: Label-Level: Classification and Action Retrieval. Instance-Level: Retrieval, Copy Detection and Ranking. For label-level tasks, VidTAB has already provided a flexible way to evaluate models. Therefore, VidEB aims to assess existing models at a finer semantic level, focusing on instance-level tasks. Although ranking tasks are common in recommendation system scenarios, they are influenced by user information and interactions, in addition to video data. Based on prior research [74], using frozen embeddings for video features does not consistently improve recommendation tasks (resulting in minimal or even negative effects). Thus, we have narrowed the final dataset scope to instance-level retrieval and copy detection. Apart from the traditional classification tasks, the evaluation of representations typically involves standard benchmarks such as video action retrieval [75, 76, 77], which primarily rely on class labels. However, this approach often overlooks the overall scene context and exhibits an overlap with recognition tasks. In contrast, inspired by previous works [78, 66, 79, 80, 81], we establish more rigorous criteria for embedding evaluation in Table 2. Specifically, we require the model to determine the priority and retrieve individual samples based on the overall similarity, rather than solely relying

on class labels. This evaluation protocol provides a more comprehensive assessment of the model's capability to encapsulate subtle visual information.

**Evaluation protocol.** To facilitate fine-grained embedding evaluation, we incorporate two tasks for assessment: **(1) Hierarchical Video Retrieval** aims to retrieve videos from a database that closely matches the query video in terms of scene, viewpoint, and temporal context. According to previous work [65], videos related to the query are categorized into three levels based on their similarity to the query: Duplicate Scene Videos (DSVs), Complementary Scene Videos (CSVs), and Incident Scene Videos (ISVs), as shown in Table 2: Consequently, the retrieval tasks are structured into three hierarchical levels: *Duplicate Scene Video Retrieval*: only DSVs are positive instances. *Complementary Scene Video Retrieval*: both DSVs and CSVs are positive instances. *Incident Scene Video Retrieval*: DSVs, CSVs, and ISVs are all positive instances. For the evaluation metric, we follow [65] to utilize the mean Average Precision (mAP) to assess the quality of video ranking. **(2) Video Copy Detection** aims to detect edited copies of the query video. Instead of the ranking/retrieval task where all video pairs need to be sorted according to video embedding similarity, it is required to identify a set of video pairs that contain edited versions of the given query. Following [66], we consider the micro-AP ($\mu$AP) as our evaluation metric that operates on all queries jointly and takes the confidence scores into account.

# 4 Benchmarking Video Foundation Models

## 4.1 Targets and details of evaluation

**Evaluation targets**   We evaluate twenty open-source vision foundation models. Including: (1) twelve video foundation models, covering ***different pre-training paradigms, model scales, and training data scales***, to analyze the impact of these factors on the generalization capability of foundation models. (2) five image foundation models to observe ***how much generalization capability trained on image data can exhibit in video understanding***. (3) three image-to-video methods based on image foundation models to assess the ***effectiveness of current efficient transfer methods***.

**Implementation details**   All models take 8 frames (16 frames if the model has temporal downsampling), with each frame being 224x224 in size as input. For VidTAB, to ensure fair comparison and efficient assessment, we train all models for the same number of epochs and made minor adjustments to the hyperparameters to ensure convergence. For VidEB, all models take 16 frames, with each frame being 224x224 in size as input. In hierarchical video retrieval, the similarity of video-level embedding determines the ranking of retrieval results. In video copy detection, each sample is segmented into 5 clips. The detection confidence score for the entire video is derived from the maximum frame-wise similarity computed for each query-reference pair. See the Appendix for more details.

## 4.2 Results on VidTAB

**Zero-shot evaluation**   To preliminarily assess the characteristics and difficulty of the dataset, we first evaluate the zero-shot performance of the eight tasks we created using two image language models and two video language models. As shown in the top section of Table 3, both image and video models demonstrated some level of performance for action-related tasks, with video models exhibiting relatively higher performance. For tasks involving low-level information understanding, such as Quality Assessment task, image models performed significantly better. In contrast, for other tasks involving scenarios typically unseen in training data, such as medical surgery videos or Safety Review tasks requiring complex semantic reasoning, all models exhibited almost no performance.

**Main results**   Table 5 presents the evaluation results on VidTAB. We summarize our findings as follows. **On the whole, (1)** Despite exhibiting a degree of generalization capability, *current vision FMs still struggle to adapt to unseen video tasks with limited training samples.* VFMs outperform IFMs, particularly in tasks related to action and behavior understanding. However, IFMs exhibit superior performance on more novel tasks, specifically in the domains of safety and quality, especially when combined with image-to-video adaptation techniques. **(2)** The *adaptation performance of models generally increases with the growth of data and model size*, as observed by the improvements observed from V-JEPA-L to V-JEPA-H (+1.5) and ViCLIP-L-10M to ViCLIP-L-200M (+1.3).

Table 5: **Evaluating state-of-the-art FMs on the VidTAB**. The best and second-best results of foundation models are noted by blue and underline, respectively. 'I', 'V', and 'IV' denote image data, video data, and mixed image-video data, respectively. Data marked in gray indicates that the model uses a model trained on that data as initialization. 'K710ft' indicates that the model was fine-tuned with supervision using the labeled action recognition dataset Kinetics-710 (0.66M). Considering the random error in few-shot experiments, we conducted 3-fold experiments for both 4-shot and 16-shot settings, and used their mean as the final result. We also provide the results of full finetuning in the appendix.

| | # Params. (M) | # Pt. Data | Average | Action | | Science | | Safety | | Quality | Emotion |
| --- | --- | --- | --- | --- | --- | --- | --- | --- | --- | --- | --- |
| | | | | Dark. | Long. | Medical. | Animal. | Harmful. | Fake. | Quality. | Emotion. |
| Random | - | - | 22.7 | 9.1 | 10.0 | 6.3 | 8.3 | 33.3 | 50.0 | 50.0 | 14.3 |
| *Zero-shot performance of visual language models* | | | | | | | | | | | |
| CLIP-L [5] | 428 | **I**-400M | 35.7 | 29.2 | 34.6 | 12.5 | 32.9 | 42.1 | 56.3 | 65.5 | 12.9 |
| EVA-CLIP-g [67] | 1365 | **I**-2B | 36.0 | 32.8 | 37.2 | 9.4 | 28.5 | 39.6 | 52.8 | 69.5 | 17.9 |
| ViCLIP-L [20] | 428 | I-400M+**V**-200M | 33.6 | 26.2 | 37.5 | 8.3 | 29.3 | 32.1 | 52.2 | 53.9 | 29.0 |
| InternVideo2$_{stage2}$ [26] | 1350 | **IV**-1.1M+**IV**-25.5M | 40.6 | 37.1 | 40.2 | 11.5 | 45.2 | 59.1 | 51.3 | 56.1 | 24.3 |
| *Image Foundation Model* | | | | | | | | | | | |
| CLIP-L [5] | 316 | **I**-400M | 43.2 | 31.9 | 37.8 | 32.3 | 37.4 | 54.2 | 58.2 | 66.6 | 27.6 |
| SigLiP-SO [82] | 444 | **I**-4.11B | 43.3 | 27.6 | 38.4 | 36.5 | 35.8 | 53.3 | 58.5 | 67.8 | 28.5 |
| EVA-g [83] | 1035 | **I**-2B | 45.8 | 40.2 | 47.1 | 34.4 | 41.0 | 51.8 | 55.2 | 68.1 | 29.0 |
| DINOv2-L [84] | 317 | **I**-142M | 42.7 | 40.8 | 45.0 | 39.6 | 36.1 | 38.9 | 52.2 | 63.2 | 25.6 |
| DINOv2-g [84] | 1165 | **I**-142M | 44.4 | 37.8 | 46.4 | 42.7 | 36.0 | 48.5 | 53.2 | 64.3 | 26.3 |
| *Image Foundation Model with **image-to-video adaptation method*** | | | | | | | | | | | |
| ST-Adapter-CLIP-L [70] | 328 | **I**-400M | 46.5 | 42.4 | 44.3 | 31.2 | 40.1 | 47.4 | 64.6 | 71.5 | 30.4 |
| AIM-CLIP-L [71] | 328 | **I**-400M | 48.8 | 41.5 | 50.0 | 38.5 | 40.2 | 46.4 | **69.5** | **73.7** | 30.6 |
| ZeroI2V-CLIP-L [72] | 303 | **I**-400M | 46.3 | 40.3 | 47.0 | 31.2 | 40.2 | 46.1 | 65.2 | 69.9 | 30.5 |
| *Video Foundation Model* | | | | | | | | | | | |
| ViCLIP-L-10M [20] | 316 | I-400M+**V**-10M | 41.8 | 31.2 | 42.7 | 30.2 | 35.3 | 47.9 | 53.9 | 66.2 | 26.9 |
| ViCLIP-L-200M [20] | 316 | I-400M+**V**-200M | 43.3 | 33.8 | 46.4 | 30.2 | 37.9 | 47.4 | 54.9 | 65.9 | 27.5 |
| VideoMAEv1-L [16] | 316 | **V**-0.24M | 43.3 | 45.6 | 30.8 | 31.2 | 37.4 | 56.5 | 51.9 | 68.7 | 24.0 |
| VideoMAEv1-H [16] | 651 | **V**-0.24M | 44.7 | 45.5 | 31.0 | 35.4 | 38.6 | 55.8 | 51.8 | 70.5 | 29.1 |
| VideoMAEv2-g [22] | 1037 | **V**-1.35M | 37.8 | 35.2 | 18.3 | 18.8 | 33.7 | 59.6 | 50.9 | 64.7 | 21.6 |
| VideoMAEv2-g$^{k710pt}$ [22] | 1037 | **V**-1.35M+K710ft | 54.0 | **76.4** | 72.6 | 50.0 | 42.4 | 43.8 | 56.9 | 63.2 | 27.0 |
| UMT-L$_{stage1}$ [21] | 316 | **V**-0.66M | 44.0 | 34.3 | 35.4 | 30.0 | 34.2 | 45.6 | 53.6 | 64.7 | 27.0 |
| UMT-L$_{stage2}$ [21] | 316 | **V**-0.66M+**IV**-25M | 44.0 | 34.2 | 43.9 | 22.9 | 39.4 | **63.9** | 53.0 | 67.3 | 27.4 |
| V-JEPA-L [23] | 318 | **V**-2M | 43.5 | 50.4 | 34.3 | 39.6 | 39.7 | 43.9 | 51.7 | 66.7 | 21.4 |
| V-JEPA-H [23] | 653 | **V**-2M | 45.1 | 53.8 | 37.6 | 35.4 | 40.4 | 47.3 | 53.0 | 68.1 | 25.1 |
| InternVideo2-1B$_{stage1}$ [26] | 1037 | **IV**-1.1M | 46.1 | 45.2 | 50.3 | 33.3 | 38.7 | 52.3 | 53.5 | 65.9 | 29.3 |
| InternVideo2-1B$_{stage1}$ [26] | 1037 | **IV**-1.1M+K710ft | **56.7** | 75.6 | **77.5** | 53.1 | 45.4 | 47.2 | 55.5 | 66.2 | **33.2** |
| InternVideo2-1B$_{stage2}$ [26] | 1037 | **IV**-1.1M+**IV**-25.5M | 53.6 | 66.0 | 71.1 | 38.5 | **50.0** | 53.6 | 54.7 | 64.3 | 30.3 |

**For the pre-training data, (3)** *While augmenting video training data is generally beneficial, it can negatively impact the performance on some tasks.* For both VideoMAEv2-g and InternVideo2-1B$_{stage1}$, fine-tuning on Kinetics-710 data significantly enhances Action-related tasks, but consistently degrades certain Safety and Quality tasks. Similar findings are observed with ViCLIP-L, where post-pretraining on a large-scale video dataset improves Action-related tasks but diminishes performance in other domains (Science, Safety, Quality, Emotion). It could be attributed to the limited diversity of the current video training data. **(4)** *For models trained on single-modal visual data, incorporating additional weak-supervised post-pretraining with visual-text data leads to significant improvements in adaptation capabilities.* This is evident in the performance gains observed from UMT-L$_{stage1}$ to UMT-L$_{stage2}$ (+3.6) and from InternVideo2-1B$_{stage1}$ to InternVideo2-1B$_{stage2}$ (+8.0). Interestingly, this finding contradicts previous conclusions drawn from commonly used action recognition benchmarks, suggesting that these benchmarks may introduce bias. **For the pre-training paradigms of model, (5)** *The effectiveness of pre-training paradigms in scaling model size might not be adequately validated on popular action recognition benchmarks.* While VideoMAEv2 successfully scaled a model to 1B parameters using the dual masking strategy [22], its adaptation performance (37.7 vs 44.4) significantly declined compared to VideoMAEv1-H. Interestingly, VideoMAEv2-g demonstrated remarkable performance after fine-tuning on Kinetics-710 (0.66M), suggesting that the abundant labeled data may have compensated for the shortcomings of its pre-training performance. **(6)** *Single-modal self-supervised pre-training paradigms exhibit superior data efficiency compared to multimodal weakly-supervised pre-training paradigms.* Notably, V-JEPA and VideoMAEv1, trained solely on relatively small-scale unlabeled video data via self-supervised pre-training, demonstrate comparable or even superior performance to ViCLIP, which is trained on a massive dataset of video-

Table 6: **Evaluation of State-of-the-Art Foundation Models on the VidEB Dataset.** "K400pt" and "K400ft" denote that the model is pre-trained and fine-tuned, respectively, using the labeled action recognition dataset Kinetics-400 (0.31M). MCL: Multi-modal Contrastive Learning, SCL: Self-supervised Contrastive Learning, MVM: Masked Video Modeling, SFT: Supervised Fine-tuning. Other notations are consistent with those in Table 5.

| | Pretrain Tasks | # Pretrain Data | Average | Scene | | | |
| --- | --- | --- | --- | --- | --- | --- | --- |
| | | | | Duplicate | Complementary | Incident | Copyright |
| *Image Foundation Model* | | | | | | | |
| CLIP-L [5] | MCL | **I**-400M | 43.0 | 41.1 | 46.4 | 52.0 | 32.3 |
| EVA-g [83] | MCL | **I**-2B | 37.1 | 41.4 | 46.1 | 51.7 | 9.3 |
| SigLiP-SO [82] | MCL | **I**-4.11B | 38.6 | 40.6 | 45.5 | 51.5 | 16.9 |
| DINOv2-L [84] | SCL | **I**-142M | 45.6 | 49.0 | 53.5 | 54.3 | 25.6 |
| DINOv2-g [84] | SCL | **I**-142M | 48.6 | **50.5** | **55.1** | **56.0** | 32.8 |
| *Video Foundation Model* | | | | | | | |
| VideoMAEv1-L [16] | MVM | K400pt | 12.9 | 14.5 | 15.1 | 13.2 | 8.8 |
| VideoMAEv1-L-K400ft [16] | MVM+SFT | K400pt+ft | 27.4 | 27.6 | 30.2 | 30.3 | 21.6 |
| VideoMAEv2-g [22] | MVM | **V**-1.35M | 11.6 | 14.8 | 15.4 | 13.4 | 2.8 |
| VideoMAEv2-g-K710ft [22] | MVM+SFT | **V**-1.35M+K710ft | 37.4 | 33.8 | 37.1 | 37.1 | 41.7 |
| UMT-L$_{stage1}$ [21] | MVM | **V**-0.66M | 41.1 | 42.2 | 46.6 | 49.6 | 25.7 |
| UMT-L$_{stage1}$-K710ft [21] | MVM+SFT | **V**-0.66M+K710ft | 29.0 | 26.4 | 29.4 | 30.3 | 30.0 |
| UMT-L$_{stage2}$ [21] | MVM+MCL | V-0.66M+**IV**-25M | 34.2 | 33.4 | 37.3 | 40.6 | 25.4 |
| V-JEPA-L [23] | MVM | **V**-2M | 19.7 | 21.3 | 23.9 | 21.7 | 12.0 |
| V-JEPA-H [23] | MVM | **V**-2M | 20.2 | 21.5 | 23.7 | 21.2 | 14.3 |
| InternVideo2-1B$_{stage1}$ [26] | MVM | **IV**-1.1M | **50.4** | 47.3 | 52.1 | 54.9 | 47.3 |
| InternVideo2-1B$_{stage1}$-K710ft [26] | MVM+SFT | **IV**-1.1M+K710ft | 33.9 | 30.5 | 34.2 | 34.1 | 36.9 |
| InternVideo2-1B$_{stage2}$ [26] | MVM+MCL | IV-1.1M+**IV**-25.5M | 34.6 | 32.4 | 36.8 | 39.9 | 29.3 |

text pairs. **In addition, (7)** *Effective adaptation method for FMs is crucial.* Three image-to-video methods based on CLIP-L achieved significant performance improvements compared to using an attentive probe directly. We believe this represents a promising avenue for future research.

### 4.3   Results on VidEB

The main results of VidEB are presented in Table 6. We evaluate the embedding performance using different pre-training paradigms for IFMs and VFMs as frozen feature extractors. Surprisingly, **IFMs performs better than most VFMs**, likely due to the existing gap in spatial modeling capabilities between VFMs and IFMs. **For the pre-training paradigms of the model, (1)** *The contrastive learning (CL) based approach consistently excels in embedding evaluation.* Due to CL's emphasis on the relationships between samples during training, DINOv2, which focuses solely on vision, outperforms vision-language contrastive methods like CLIP across multiple tasks. **(2)** *The effectiveness of masked video modeling is closely tied to the targets it reconstructs or aligns with.*   With higher semantic richness, it shows progressive improvements in embedding quality for VideoMAE-L, V-JEPA-L, and UMT-L$_{stage1}$. **(3)** *Vision-centric pretraining outperforms Multi-modal pretraining in vision-centric scenarios.* Comparing UMT-L$_{stage1}$ and InternVideo2-1B$_{stage1}$ with their multi-modal counterparts UMT-L$_{stage2}$ and InternVideo2-1B$_{stage2}$, the introduction of visual-text pair data in multi-stage training does not enhance performance in vision-centric scenarios. This is also consistent with the performance differences observed between DINO and CLIP-style pre-training methods. Additionally, we assess the **impact of fine-tuning on the embedding evaluation of these pre-trained models**. **(4)** *Labels bring new semantic information or disrupt existing finer-grained semantic information.* The performance variations after fine-tuning differ based on the pre-training strategy. For UMT-L$_{stage1}$ and InternVideo2-1B$_{stage1}$, fine-tuning leads to a significant drop in performance (-12.1 for UMT and -16.5 for InternVideo) due to the introduction of more singular label information, which causes catastrophic forgetting. In contrast, VideoMAE and VideoMAEv2 show substantial performance gains (+14.5 and +25.8, respectively) because the low-level semantics learned during pre-training are less abstract and benefit more from the addition of high-level label information.

## 5   Conclusions

We present VideoEval, a comprehensive benchmark suite for efficiently evaluating the VFMs. To this end, we establish VidTAB, which explores suitable evaluation tasks and protocols for VFMs from the perspective of assessing their adaptability to unknown tasks with limited samples. Additionally, we create VidEB to evaluate the capability of VFMs' feature embedding in directly supporting downstream tasks. Utilizing VideoEval, we conduct a large-scale study involving 20 popular open-source vision foundation models, providing valuable insights for future research directions.

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
