

Figure 1: **Examples of VidTAB.**

In this appendix, we provide more details of VideoEval from the following aspects:

- Details of our benchmark are in § 1.
- Details of training and evaluation, can be found in § 2.
- Ethics etatement of the datasets are in § 3
- Limitations and potential negative societal impacts are in § 4

# 1 Details of Benchmark

**Examples of VidTAB**   As shown in Figure 1, we present some examples of tasks in VidTAB.

**Details of VidTAB**   The detals of task construction are presented in Table 1. For each category in one task, we sample 4, 16, and 100 samples, respectively. Given the limited volume of medical surgery data, we only sample 4 samples from each category for few-shot evaluation. To mitigate the impact of randomness, we sampled two sets of data for four tasks and obtained the benchmark results. We found that the randomness of sampling had negligible effects on the final rankings of VFMs in the benchmark.

# 2 Details of Training and Evaluation

**Checkpoints of Evaluation Models**   We provide checkpoints of the models we evaluate for reproducibility of our results.

Table 1: **Task details of VidTAB.** All videos are collected from the public datasets for building tasks of VidTAB.

| Task | Source | Num. test sample | Details of Task Construction |
|---|---|---|---|
| Action Recognition | ARID [1] | 2011 | We directly employ the original classification task definition. Specifically, 11 categories. |
| Action Recognition | BreakFast [2] | 822 | We directly employ the original classification task definition. Specifically, 12 categories. |
| Medical Surgery | SurgicalActions160 [3] | 96 | We directly employ the original classification task definition. Specifically, 16 categories. |
| Animal Behavior | Animal Kingdom [4] | 2268 | Since the annotations in this dataset included multiple labels, we filtered out all categories with only single labels and then selected categories with more than 150 samples. This resulted in a final set of 12 categories. |
| Fake Face | FaceForensics++ [5] | 1800 | We used the original 1000 videos as positive samples. Then, we divided the original videos into five parts and used the Deepfakes, Face2Face, FaceShifter, FaceSwap, and NeuralTextures methods to generate 1000 negative samples by face-swapping. We then selected 1800 of these samples as the test set and the remaining as the training set. |
| Harmful Content | mob [6] | 1661 | We categorized videos into three classes based on their content: those containing fast repetitive movements and violence activities, those containing unpleasant appearances and obscene scenes, and those containing no malicious information at all. This resulted in a three-class classification task. |
| Quality Assess | DOVER [7] | 724 | To convert the task into a classification problem, we sorted the "overall score" label and divided the videos into positive and negative samples, with the top and bottom 40% constituting the respective categories. |
| Emotion Analysis | CAER [8] | 3953 | We directly employ the original classification task definition. Specifically, 7 categories. |

- CLIP [9]: `https://huggingface.co/openai/clip-vit-large-patch14`

- EVA-CLIP [9]: `https://huggingface.co/QuanSun/EVA-CLIP`

- ViCLIP [10]: `https://github.com/OpenGVLab/InternVideo/tree/main/Data/InternVid`

- InternVideo2 [11]: `https://huggingface.co/collections/OpenGVLab/internvideo2-6618ccb574bd2f91410df5cd`

- SigLiP [12]: `https://huggingface.co/google/siglip-so400m-patch14-384`

- DINOv2 [13]: `https://huggingface.co/facebook/dinov2-giant`

- VideoMAE [14]: `https://github.com/MCG-NJU/VideoMAE/blob/main/MODEL_ZOO.md`

- VideoMAEv2 [15]: `https://github.com/OpenGVLab/VideoMAEv2/blob/master/docs/MODEL_ZOO.md`

- UMT [16]: `https://github.com/OpenGVLab/unmasked_teacher`

- V-JEPA [17]: `https://github.com/facebookresearch/jepa`

**Trainging strategies** Specific hyperparameter configurations are available in the configs provided in our code repository. In essence, we train all models for 25 epochs using a similar training strategy, employing the Adam optimizer, a learning rate of 5e-5, and only utilizing RandomResizedCrop for data augmentation. And we use a single clip to obtain the final evaluation performance.

**Total amount of compute and the type of resources used** Leveraging the low cost of our evaluation protocol, we conducted each experiment involving a single VFM and a single task on one A100-80G GPU. We performed approximately 300 such experiments, each taking around 1-2 hours, resulting in a total of around 400 GPU hours.

## 3 Ethics Statement

**license of the datasets** The dataset we are using is collected from publicly accessible sources, all licensed under Creative Commons (CC-BY) or other open-source licenses. We have diligently followed all legal requirements to integrate this data into our research, emphasizing the importance of transparency in data licensing for proper attribution and appropriate use. Although we have taken steps to ensure the inclusion of suitable content, we recognize that some problematic content may still exist. If you encounter any such content, please notify us immediately so we can take corrective action to maintain a dataset free from inappropriate material. We are dedicated to maintaining a high-quality, ethically responsible dataset and pledge to uphold principles of privacy and transparency in all our work.

**Privacy or safety concerns in video** For personally identifiable information or offensive content in video, our data collection sources have been carefully considered, and we believe these issues are not present. However, if you discover any oversights, please do not hesitate to contact us promptly.

Table 2: **Evaluating state-of-the-art VFMs on the VidTAB with Full Finetuning**. The best and second-best results of foundation models are noted by `blue` and underline, respectively. We present the results in the form of '4s/16s/100s,' representing the outcomes of 4-shot, 16-shot, and 100-shot experiments.

| | | Action | | Science | | Safety | | Quality | Emotion |
|---|---|---|---|---|---|---|---|---|---|
| | Average | Dark Scene | Long Video | Medical Surgery | Animal Behavior | Harmful Content | Fake Face | Quality Assess | Emotion Analysis |
| Random | 22.7 | 9.1 | 10.0 | 6.2 | 8.3 | 33.3 | 50.0 | 50.0 | 14.3 |
| *Video Foundation Model* | | | | | | | | | |
| ViCLIP-L-10M [10] | 37.9 | 22.6/18.9/29.5 | 16.4/24.8/45.7 | 30.2 | 26.3/29.7/41.5 | 35.1/38.2/54.2 | 51.2/50.8/53.7 | 56.9/65.9/72.5 | 20.3/17.2/32.7 |
| ViCLIP-L-200M [10] | 38.3 | 21.1/20.5/37.2 | 13.6/21.2/53.0 | 30.2 | 25.1/30.6/43.6 | 36.6/40.2/46.8 | 50.4/51.5/53.7 | 57.2/67.7/71.6 | 19.8/19.7/32.2 |
| VideoMAEv1-H [14] | 34.0 | 12.8/13.5/72.1 | 9.6/10.0/36.7 | 39.6 | 18.5/22.0/47.8 | 32.5/33.1/37.2 | 50.3/50.3/50.7 | 44.2/50.8/66.6 | 15.2/14.3/19.0 |
| VideoMAEv2-g [15] | 34.0 | 13.1/13.4/76.1 | 31.4/12.3/34.3 | 18.8 | 12.2/18.7/50.8 | 29.4/30.2/41.5 | 50.8/50.6/50.6 | 52.0/55.3/62.2 | 12.7/14.2/17.4 |
| VideoMAEv2-g$^{k710pt}$ [15] | 48.6 | 30.4/77.3/ **94.0** | 31.2/52.9/89.0 | 57.3 | 12.6/32.0/64.5 | 33.1/39.4/41.8 | 49.8/50.4/54.7 | 54.3/59.8/71.4 | 16.6/17.2/39.3 |
| V-JEPA-L [17] | 49.2 | 43.2/78.8/88.5 | 25.2/52.0/86.0 | 46.9 | 26.6/37.1/59.9 | 38.5/36.0/46.4 | 50.2/50.8/55.9 | 54.3/68.0/76.9 | 15.0/17.9/27.4 |
| V-JEPA-H [17] | 52.5 | 45.2/ **80.7** /90.8 | 24.7/48.5/87.1 | 46.9 | 26.7/38.1/60.6 | 40.4/41.7/ **58.5** | 50.4/51.2/68.2 | 59.8/ **71.3** / **79.3** | 20.9/20.4/43.4 |
| InternVideo2-1B$_{stage1}$ [11] | 52.1 | 20.3/56.0/80.6 | 27.7/70.0/92.5 | 66.7 | 27.2/38.2/58.8 | 41.5/36.0/50.0 | 52.6/52.4/75.0 | **60.9** /69.0/77.8 | 16.1/31.8/45.4 |
| InternVideo2-1B$_{stage1}^{k710pt}$ [11] | **59.4** | **59.5** /79.9/88.9 | **60.8** / **82.6** / **95.6** | **71.9** | 31.7/ **46.4** / **68.0** | 44.0/37.7/50.1 | **53.3** / **53.9** / **83.2** | 59.4/65.4/77.9 | 22.9/28.0/ **45.8** |
| InternVideo2-1B$_{stage2}$ [11] | 59.0 | 55.1/75.6/89.3 | 55.4/77.7/93.7 | 60.4 | **33.3** / 51.0 /67.7 | **54.2** / 42.2 /55.1 | 50.9/53.4/76.9 | 58.5/67.0/77.4 | **23.9** / **34.6** /44.1 |

# 4 Limtiations and Societal Impacts

**Limitations**  Firstly, due to the limitations of diversity and accuracy in our video sources and annotations, which were gathered from public resources, we plan to further enrich the task in the future by incorporating manual annotations and self-collected data. Secondly, considering the evaluation cost and simplicity, we currently only evaluate tasks like classification and retrieval, which primarily rely on VFMs' global information extraction capabilities. We have not yet considered tasks like spatio-temporal action detection and temporal grounding, which assess other aspects of VFMs' capabilities. We will expand the scope of evaluation in the future.

**Potential negative societal impacts**  While our evaluation includes tasks like synthetic video recognition and harmful information recognition, these serve only as indicators of the model's overall performance in this area and cannot be used to accurately evaluate the actual performance of a specific task. If researchers or engineers in society attempt to use VFMs to perform these specific tasks, our benchmark can serve as a reference for their choice of VFMs, but it cannot be used as the final standard for evaluating that task. Otherwise, it may have negative impacts on the corresponding real-world applications.

Table 3: **Evaluating state-of-the-art FMs on the VidTAB with Attentive Probe**. The best and second-best results of foundation models are noted by blue and underline, respectively. We present the results in the form of '4s/16s/100s,' representing the outcomes of 4-shot, 16-shot, and 100-shot experiments.

| | | Action | | Science | | Safety | | Quality | Emotion |
| | Average | Dark Scene | Long Video | Medical Surgery | Animal Behavior | Harmful Content | Fake Face | Quality Assess | Emotion Analysis |
|---|---|---|---|---|---|---|---|---|---|
| Random | 22.7 | 9.1 | 10.0 | 6.2 | 8.3 | 33.3 | 50.0 | 50.0 | 14.3 |
| *Image Foundation Model* | | | | | | | | | |
| CLIP-L [9] | 44.3 | 20.5/21.6/53.5 | 15.1/21.8/76.6 | 32.3 | 29.5/36.3/46.4 | 49.5/48.1/65.1 | 52.8/57.1/64.6 | 60.2/69.4/70.3 | 21.8/22.8/38.2 |
| SigLiP-SO [12] | 43.9 | 20.1/23.7/39.0 | 16.8/27.5/71.0 | 36.5 | 25.0/35.0/47.4 | 49.8/48.1/62.1 | 54.8/57.2/63.4 | 58.8/68.9/75.7 | 21.7/23.4/40.5 |
| EVA-g [18] | 46.9 | 26.8/33.5/60.3 | 22.1/36.7/82.5 | 34.4 | 31.9/39.9/51.3 | 49.6/45.5/60.4 | 51.6/55.1/58.8 | 60.4/69.3/74.6 | 23.2/24.2/39.7 |
| DINOv2-L [13] | 42.9 | 26.2/37.3/58.9 | 17.0/37.1/80.8 | 39.6 | 26.5/36.3/45.4 | 37.1/31.6/48.0 | 51.0/52.0/53.6 | 54.7/64.5/70.3 | 21.8/22.5/32.4 |
| DINOv2-g [13] | 44.5 | 23.7/33.7/56.1 | 17.7/38.4/83.2 | 42.7 | 26.6/36.1/45.4 | 40.9/44.2/60.3 | 51.8/51.8/55.9 | 54.5/65.5/72.8 | 21.8/22.7/34.4 |
| *Image Foundation Model with **image-to-video adaptation method*** | | | | | | | | | |
| ST-Adapter-CLIP-L [19] | 47.9 | 21.2/37.3/68.6 | 17.4/35.1/80.5 | 31.2 | 30.1/39.6/50.7 | 48.0/42.9/51.4 | 53.3/59.6/80.8 | 62.4/71.9/80.1 | 20.3/22.1/48.7 |
| AIM-CLIP-L [20] | 49.7 | 22.4/39.3/62.8 | 21.2/47.5/81.3 | 38.5 | 29.7/39.1/51.7 | 44.3/38.9/55.9 | **57.4 / 67.2 / 83.8** | **64.9 / 73.0 / 83.2** | 21.8/24.7/45.2 |
| ZeroI2V-CLIP-L [21] | 47.6 | 22.2/37.8/61.0 | 21.2/40.6/79.1 | 31.2 | 31.0/39.4/50.1 | 40.9/37.9/59.5 | 55.5/57.7/82.3 | 58.8/70.4/80.4 | 20.1/22.6/ **48.7** |
| *Video Foundation Model* | | | | | | | | | |
| ViCLIP-L-10M [10] | 42.8 | 22.4/25.2/46.1 | 19.6/35.3/73.2 | 30.2 | 26.3/34.4/45.2 | 38.0/46.9/58.8 | 51.6/53.4/56.8 | 59.5/68.0/71.0 | 21.2/22.4/37.1 |
| ViCLIP-L-200M [10] | 44.5 | 25.9/32.4/56.2 | 21.1/38.0/74.7 | 30.2 | 28.2/37.0/48.6 | 45.8/44.6/51.8 | 52.5/53.6/58.7 | 56.4/70.2/71.1 | 21.0/23.2/38.2 |
| VideoMAEv1-L [14] | 44.4 | 19.0/35.1/82.6 | 12.8/14.6/65.1 | 31.2 | 25.6/31.1/55.4 | 62.1/49.6/57.9 | 50.5/51.1/54.2 | 57.9/70.3/77.9 | 18.9/16.7/36.5 |
| VideoMAEv1-H [14] | 45.6 | 17.7/35.4/83.4 | 11.8/15.7/65.6 | 35.4 | 24.8/32.8/58.2 | 56.0/45.6/ **65.9** | 50.4/51.3/53.6 | 62.6/70.4/78.4 | **26.1** /26.4/34.8 |
| VideoMAEv2-g [15] | 39.6 | 15.9/19.6/70.0 | 14.2/14.0/26.8 | 18.8 | 25.2/26.1/49.7 | 63.1/52.9/62.7 | 50.9/50.5/51.2 | 56.5/62.6/74.9 | 16.7/21.9/26.2 |
| VideoMAEv2-g$^{k710pt}$ [15] | 54.4 | 63.4/76.9/ **88.8** | 59.5/75.3/83.0 | 50.0 | 26.5/41.3/59.3 | 41.0/41.4/49.1 | 52.9/55.2/62.6 | 52.3/65.3/72.1 | 21.4/23.1/36.6 |
| UMT-L$_{stage1}$ [16] | 41.6 | 25.5/21.8/55.5 | 14.8/22.4/68.9 | 30.0 | 24.9/32.8/44.8 | 42.4/41.4/53.0 | 51.1/52.9/56.9 | 59.3/66.3/68.5 | 24.2/20.0/36.9 |
| UMT-L$_{stage2}$ [16] | 45.9 | 25.2/26.6/50.8 | 23.6/35.2/72.8 | 22.9 | 29.6/35.4/53.3 | **66.6 / 61.8** /63.3 | 50.6/51.4/56.9 | 58.9/68.5/74.4 | 25.0/20.5/36.6 |
| V-JEPA-L [17] | 43.8 | 26.8/46.7/77.8 | 18.1/27.5/57.4 | 39.6 | 28.0/36.0/55.2 | 37.1/41.3/53.2 | 50.9/50.9/53.4 | 55.2/67.6/77.2 | 18.5/17.8/27.8 |
| V-JEPA-H [17] | 46.0 | 28.1/47.5/85.7 | 17.2/26.9/68.6 | 35.4 | 27.6/36.6/57.0 | 40.4/42.0/59.6 | 51.3/52.5/55.3 | 58.0/68.4/77.9 | 22.1/20.3/32.9 |
| InternVideo2-1B$_{stage1}$ [11] | 47.2 | 27.4/38.5/69.7 | 22.2/42.5/86.1 | 33.3 | 28.5/36.3/51.3 | 44.7/48.2/64.1 | 50.8/53.0/56.8 | 57.6/67.1/73.1 | 23.0/24.0/40.9 |
| InternVideo2-1B$_{stage1}^{k710pt}$ [11] | **57.0** | **66.4 / 77.9** /82.4 | **65.3 / 77.5 / 89.8** | **53.1** | 31.3/44.1/60.7 | 43.9/42.4/55.4 | 51.9/54.7/59.9 | 57.1/66.5/75.0 | 23.3/ **33.3** /43.0 |
| InternVideo2-1B$_{stage2}$ [11] | 54.9 | 54.4/66.6/76.9 | 56.0/71.7/85.6 | 38.5 | **37.2 / 50.4 / 62.5** | 51.0/46.2/63.6 | 51.6/54.4/58.2 | 53.9/65.8/73.2 | 21.8/29.3/39.9 |