# OpenReview forum: "VideoEval: Comprehensive Benchmark Suite for Low-Cost Evaluation of Video Foundation Model"
_NeurIPS.cc/2025/Datasets_and_Benchmarks_Track — Submitted to NeurIPS 2025 Datasets and Benchmarks Track_

### Official Review · Reviewer_6mG5 · 2025-06-28

**Rating:** 5
**Confidence:** 4

**Summary:**

The paper introduces VideoEval, a comprehensive and low-cost benchmark suite for evaluating Video Foundation Models (VFMs). Addressing the key limitations of current VFM benchmarks—such as limited task diversity, high evaluation costs, saturated performance metrics, and biases introduced by language models—the paper proposes two sub-benchmarks: (1) VidTAB (Video Task Adaptation Benchmark): Evaluates model adaptability to unseen tasks under few-shot conditions. (2) VidEB (Video Embedding Benchmark): Measures the utility of feature embeddings for downstream tasks. The paper conducts a large-scale empirical study on 20 vision foundation models and reports several novel insights, such as the limited generalization of current VFMs and the overestimated efficacy of some pretraining paradigms.

**Additional Feedback:**

1. Line 36 “In image realm, Previous works”
2. Line 39 “previous works primarily evaluates”
3. Some error in Table 2

**Dataset Code Accessibility:**

Yes

**Dataset Code Comments:**

The dataset and code are well-documented, publicly available, and easy to access, ensuring reproducibility and ease of use.

**Ethical Comments:**

This paper’s research is a continuation of a series of previous studies, and therefore the benchmark does not raise many ethical concerns from my perspective.

**Ethical Considerations:**

No, there are no or only very minor ethics concerns

**Final Justification:**

Most of my concerns have been addressed in the author’s responses, and the implemented changes are likely to improve the paper’s final quality. My initial score remains the same.

**Limitations Weaknesses:**

1. Certain tasks, such as animal behavior analysis and surgical procedures, are less frequently encountered in commercial or industrial environments, potentially limiting the benchmark's broad applicability.

2. The benchmark primarily focuses on vision-based evaluation and does not incorporate recent advances in multimodal assessment. This limitation prevents it from fully reflecting the capabilities of models designed for cross-modal understanding. The author might consider elaborating on the interplay between vision-centric and multimodal evaluations to clarify this relationship.

**Strengths Contributions:**

1. The paper proposes a practical and efficient evaluation framework that significantly lowers the cost of assessing video foundation models. By leveraging few-shot and training-free protocols, it enables scalable comparisons without requiring extensive computational resources.

2. It expands the scope of evaluation beyond traditional action recognition, incorporating tasks from diverse domains such as medical video analysis, content moderation, quality assessment, and emotion recognition. This broader coverage better reflects the real-world use cases of VFMs.

3. Instead of relying on conventional benchmarks that may no longer differentiate model performance, the paper builds challenging and well-curated tasks designed to reveal meaningful differences in generalization ability across models.

4. The large-scale empirical study compares 20 foundation models using consistent and fair protocols. It yields insightful findings—such as the limited benefit of more training data in some cases and the surprising strength of various pretraining patterns—which are valuable for guiding future model design.

---

> ### Author Rebuttal · Authors · 2025-07-30
>
> **Q1. Certain tasks  are less frequently encountered in commercial or industrial environments, potentially limiting the benchmark's broad applicability.**
>
> We agree with your point that some of these tasks may not occur in commercial or industrial environments; however, we still consider them meaningful tasks (e.g., for aiding scientific research, among others). Additionally, due to their "out-of-domain" nature, these tasks are highly suitable for evaluating the generalization performance of VFMs.
>
> **Q2. The author might consider elaborating on the interplay between vision-centric and multimodal evaluations to clarify this relationship.**
>
>
> We strongly agree with your perspective that exploring VFMs and foundation models of other modalities (e.g., Large Language Models, LLMs) is of great importance and necessity. First, we would like to highlight the key differences between single-modal (vision-centric) and multi-modal evaluations, taking interaction with the language modality as an example:
>
> 1. For single-modal (vision-centric) evaluation, since the task heads used by all VFMs are identical and very lightweight, the evaluation results is **solely determined by the VFMs themselves**.
> 2. For multi-modal evaluation, however, the involvement of language models complicates the assessment of VFMs, with two main approaches:
> 	- Using language models that are originally matched with the VFMs to be evaluated (e.g., using CLIP-Text, which corresponds to CLIP-ViT). A notable issue here is **unfair comparison**, as different language models can obviously affect the final evaluation results. For instance, in LiT[1], significant improvements can still be achieved by freezing the ViT while only training the text encoder.
> 	- The other approach is adopting the same language model, i.e., integrating different VFMs with the same language model. For example, integrating with text encoder like BERT[2][3] for contrastive learning training to evaluate their cross-modal retrieval capabilities, or integrating with large language models to assess their performance in generative tasks such as question answering. This approach has **two major drawbacks**: (1) Excessively high training costs, as sufficient data and training time are required to converge to reasonable results; (2) Introducing overly powerful language models may bias the evaluation of VFMs. For example, consider VFMs A and B: although A has stronger temporal modeling capabilities than B, when performing multi-modal generative tasks like video captioning, using an LLM with strong temporal modeling as the text decoder may mask A’s advantages over B, thereby biasing the assessment of the VFMs’ inherent capabilities.
>
> In summary, given that our goal is centered on evaluating the VFMs themselves, our VideoEval focuses exclusively on single-modal evaluation. We believe it is complementary to multi-modal evaluation, and a more comprehensive assessment can be achieved by combining it with existing multi-modal evaluation methods.
>
> **Q3. Regarding issues of writing and presentation.**
>
> We thank you for the constructive feedback and will refine the manuscript accordingly in the next revision.
>
> [1] LiT: Zero-Shot Transfer with Locked-image text Tuning
>
> [2] BERT: Pre-training of Deep Bidirectional Transformers for Language Understanding
>
> [3] VindLU: A Recipe for Effective Video-and-Language Pretraining

---

> ### Comment · Area_Chair_7w7c · 2025-08-04
>
> Dear 6mG5,
>
> Thank you for already reading the rebuttal and the other reviews. Please make sure to engage in an open exchange with the authors asap so there is time for back and forth discussion.
>
> Best, AC

---

### Official Review · Reviewer_Hwmv · 2025-06-30

**Rating:** 3
**Confidence:** 5

**Summary:**

This work proposes VideoEval to comprehensively evaluate video foundation models (VFM). They establish the Video Task Adaption Benchmark (VidTAB) and the Video Embedding Benchmark (VidEB) from two perspectives: evaluating the task adaptability of VFMs under few-shot conditions and assessing their feature embedding’s direct applicability to downstream tasks. Based on the benchmark, they conduct analysis on 20 popular open-sourced models and provide their findings & insights.

**Dataset Code Accessibility:**

Yes

**Ethical Considerations:**

No, there are no or only very minor ethics concerns

**Final Justification:**

The authors addressed the majority of my concerns. My only concern now is about the scientific writing and would like to see a through and careful revision of the paper.

**Limitations Weaknesses:**

1. It would be helpful to explain the motivations behind selecting the five application scenarios listed in Lines 109–119. Why were these particular scenarios chosen, and how do they relate to other potential use cases? Additionally, many existing benchmarks evaluate zero-shot performance (e.g., zero-shot text-video retrieval (T-to-V or V-to-T on MSR-VTT LSMDC DiDeMo MSVD ActivityNet), which is more cost-effective as it does not require model training. It would strengthen the paper to include a comparison with such zero-shot benchmarks.
2. The paper is challenging to read and follow, especially in the sections describing the proposed design. Although the authors present numerous arguments, many are unsupported by references or empirical evidence. Section 3, in particular, requires significant revision to improve clarity and provide adequate justification.
3. It would be helpful to include more comparisons with existing benchmarks to better highlight the strengths of the proposed approach. Additionally, has any human evaluation been conducted on the benchmark? Incorporating such an evaluation would provide stronger support for the paper’s claims.
4. There are numerous typos throughout the paper, which significantly detract from its readability and do not meet the standards expected of submissions to a top-tier conference. I strongly recommend that the authors conduct at least one thorough proofreading before submitting. Due to time constraints, I cannot list all the typographical errors, but their frequency suggests a lack of care in preparation. I urge the community to take submission quality more seriously out of respect for the reviewers' time and effort.

a. L36, Previous -> previous

b. L39, evaluates -> evaluate

c. Table 2 is hard to read.

d. L109-119 is hard to understand, e.g., "recognizing actions within special scenarios", which specific scenario is being referred to here?

e. L152, "most popular benchmarks have large training sample sizes", which benchmarks? Reference here would be helpful.

f. L206, ":" -> "."

g. L236. "demonstrated some level of performance for action-related tasks" -> what does "some level" mean? good or bad?

h. L139, "We observed that both zero-shot testing and few-shot experiments based on current VFMs show that when the number of categories is too high, models often perform no better than random guessing." Are there any results (e.g., tables or figures) that support this argument? Additionally, the phrase "too high" is vague. Could you clarify what it means and to what extent it is considered "too high"?

**Strengths Contributions:**

The motivation of this work is helpful to the community: we need to find a comprehensive but cost-effective way to evaluate our video foundation models.

---

> ### Author Rebuttal · Authors · 2025-07-30
>
> **Q1. It would be helpful to explain the motivations behind selecting the five application scenarios listed in Lines 109–119. Why were these particular scenarios chosen, and how do they relate to other potential use cases?**
>
> We conducted a months-long investigation, integrating insights from multiple researchers and engineers in the field of video understanding, and selected evaluation tasks by leveraging available high-quality public video data. Our selection was guided by two core principles:
>
> 1. The selected tasks are **meaningful real-world tasks**; the practical application value of these tasks ensures the validity of the capability dimensions we evaluate.
> 2. The selected tasks are often **unaddressed in previous evaluations**, thereby ensuring our assessment of the out-of-domain generalization of Video Foundation Models (VFMs).
>
> We elaborate below on the application value and evaluative significance of each scenario in our assessment:
>
> - Action: While numerous previous benchmarks have evaluated action recognition tasks, we still include it as part of our assessment given its status as one of the most critical video understanding tasks. We focus on specialized scenarios that have received relatively little attention in traditional action recognition tasks, specifically selecting action recognition in dark scenes and long-video scenarios. This aims to examine the action recognition performance of VFMs in relatively out-of-domain test scenarios (as most training data for existing VFMs consists of short videos in relatively bright scenes).
> - Science: "AI for Science" has emerged as a topic of growing interest in recent years but has often been overlooked in previous video understanding benchmarks. We chose animal and medical video understanding for evaluation in this domain, with specific application scenarios including animal behavior monitoring, behavioral surveillance during medical surgeries, and safety inspections.
> - Safety: Using AI for video content moderation is a key application of video understanding, encompassing tasks such as detection of AI-generated content and detection of harmful information. These tasks require models to possess more fine-grained video understanding capabilities.
> - Quality: Low-level evaluation of video quality has also been generally neglected in previous VFM assessments. We argue that a sufficiently general VFM representation should not only support high-level semantic understanding but also enable low-level visual understanding.
> - Emotion: Understanding human emotions is a crucial capability for AI models. Human emotions are complex and often require capturing microexpressions and subtle movements for accurate recognition, making this an important application scenario for VFMs.
>
> **Q2. Many existing benchmarks evaluate zero-shot performance (e.g., zero-shot text-video retrieval (T-to-V or V-to-T on MSR-VTT LSMDC DiDeMo MSVD ActivityNet), which is more cost-effective as it does not require model training. It would strengthen the paper to include a comparison with such zero-shot benchmarks.**
>
> It is important to note that we have designed a **vision-centric evaluation method specifically for Video Foundation Models (VFMs). Low cost is a key design focus of ours, but not the sole criterion**. The video-text benchmarks you mentioned, such as MSR-VTT, fall under the category of evaluation benchmarks for video-language models. While they can, to a certain extent, reflect the capabilities of VFMs, we would like to highlight the key differences between single-modal (vision-centric) and multi-modal evaluations, taking interaction with the language modality as an example:
> 1. For single-modal (vision-centric) evaluation, since the task heads used by all VFMs are identical and very lightweight, the evaluation results is **solely determined by the VFMs themselves**.
> 2. For multi-modal evaluation, however, the involvement of language models complicates the assessment of VFMs, with two main approaches:
>   - Using language models that are originally matched with the VFMs to be evaluated (e.g., using CLIP-Text, which corresponds to CLIP-ViT). A notable issue here is **unfair comparison**, as different language models can obviously affect the final evaluation results. For instance, in LiT[1], significant improvements can still be achieved by freezing the ViT while only training the text encoder.
>   - The other approach is adopting the same language model, i.e., integrating different VFMs with the same language model. For example, integrating with BERT[2] for contrastive learning training[3] to evaluate their cross-modal retrieval capabilities, or integrating with large language models to assess their performance in generative tasks such as question answering. This approach has **two major drawbacks**: (1) Excessively high training costs, as sufficient data and training time are required to converge to reasonable results; (2) Introducing overly powerful language models may bias the evaluation of VFMs. For example, consider VFMs A and B: although A has stronger temporal modeling capabilities than B, when performing multi-modal generative tasks like video captioning, using an LLM with strong temporal modeling as the text decoder may mask A’s advantages over B, thereby biasing the assessment of the VFMs’ inherent capabilities.
>
> In summary, given that our goal is centered on evaluating the VFMs themselves, our VideoEval focuses exclusively on single-modal evaluation. We believe it is complementary to multi-modal evaluation, and a more comprehensive assessment can be achieved by combining it with existing multi-modal evaluation methods.
>
> **Q3. It would be helpful to include more comparisons with existing benchmarks to better highlight the strengths of the proposed approach.**
>
> We have conducted a thorough comparison with existing benchmarks in Table 1 of the main paper. Compared with previous benchmarks of VFMs, we have defined more out-of-domain task types and more efficient evaluation protocols.
>
> **Q4. Additionally, has any human evaluation been conducted on the benchmark? Incorporating such an evaluation would provide stronger support for the paper’s claims.**
>
> We agree with your point regarding the importance of human evaluation. Given the large size of the test set (20,497 samples), it is impractical for us to conduct a full manual inspection of all samples. We therefore sample approximately 50 examples per task (balanced across classes) and evaluate them by a human who is unaware of the task background and lacks some of the knowledge required for the task, we additionally include Gemini 2.5 Flash and Gemini 2.5 Pro[4] for comparison. The results are reported as follows:
>
> | Model           | Date  | AR in Dark Scene | AR in Long Video | Medical Surgery | Animal Behavior | Fake Face | Harmfull Content | Quality Assess | Emotion Analysis |
> | --------------- | ----- | ---------------- | ---------------- | --------------- | --------------- | --------- | ---------------- | -------------- | ---------------- |
> | random          | -     | 9.1              | 10.0             | 6.3             | 8.3             | 50.0      | 33.3             | 50.0           | 14.3             |
> | human           | -     | 75.0             | 85.0             | 58.3            | 70.8            | 87.5      | 66.7             | 81.3           | 53.6             |
> | Gemini2.5-Flash | 25.06 | 34.1             | 70.0             | 22.9            | 52.1            | 62.5      | 41.7             | 71.9           | 42.9             |
> | Gemini2.5-Pro   | 25.06 | 43.2             | 70.0             | 28.1            | 56.3            | 65.6      | 50.0             | 59.4           | 46.4             |
> > Note that for human evaluation, we only asked the evaluators to judge the videos based on category names, so the accuracy may be reduced due to their lack of relevant background knowledge. Regarding the reliability of the original annotations, since the annotation information we used is either constructed from real information or manually annotated, and we have conducted manual sampling checks on these annotations during task selection, the reliability of their quality is ensured.
>
> As shown in the table above, for a human who is unaware of the task background and lacks some of the knowledge required for the task, our task remains highly challenging. And the current state-of-the-art video-understanding models still fall far short of solving our tasks.
>
>
>
>
> **Q5. Regarding issues of writing and presentation.**
>
> We sincerely apologize for the confusion caused by our writing and will undertake a thorough revision in the next version to address all the issues you have identified.
>
> [1] LiT: Zero-Shot Transfer with Locked-image text Tuning
>
> [2] BERT: Pre-training of Deep Bidirectional Transformers for Language Understanding
>
> [3] VindLU: A Recipe for Effective Video-and-Language Pretraining
>
> [4] Gemini 2.5: Pushing the frontier with advanced reasoning, multimodality, long context, and next generation agentic capabilities

---

> > ### Comment · Reviewer_Hwmv · 2025-08-05
> >
> > Thanks for the rebuttal. It addressed my major concerns. Therefore, I would like to increase my rating. However, a thorough and careful revision is required for this work considering the number of typos reviewers catch.

---

> > > ### Author Response · Authors · 2025-08-06
> > >
> > > Thank you for your reply and valuable suggestions. We will incorporate the additional analysis and explanations you asked us to provide, and will conduct a thorough re-polishing of the writing structure and content of the paper.

---

> ### Comment · Area_Chair_7w7c · 2025-08-04
>
> Dear Hwmv,
>
> Please make sure to read the other reviews and the author response and engage in an open exchange with the authors asap so there is time for back and forth discussion.
>
> Best, AC

---

### Official Review · Reviewer_e8Pz · 2025-07-02

**Rating:** 4
**Confidence:** 4

**Summary:**

This paper presents a new evaluation framework tailored to video foundation models. It introduces two complementary benchmarks: Video Task Adaptation Benchmark, which measures how well VFMs adapt to eight diverse downstream video tasks under few-shot settings. Video Embedding Benchmark, which quantifies the direct utility of VFM feature embeddings via four embedding-based tasks at different granularities.

**Additional Feedback:**

See weaknesses.

**Dataset Code Accessibility:**

Partly

**Dataset Code Comments:**

See weaknesses point 3.

**Ethical Comments:**

The author built VideoEval based on existing public datasets

**Ethical Considerations:**

No, there are no or only very minor ethics concerns

**Final Justification:**

Most concerns were addressed, but scope limitations (e.g., missing generative/multimodal tasks) and minor issues remain, so I keep my borderline accept score.

**Limitations Weaknesses:**

- Garbled characters appear in Table 2.
- Although VidTAB spans eight tasks, it omits spatio-temporal localization and generative video tasks (e.g. video QA, summarization). These are important use-cases for VFMs.
- Although the paper provides detailed code for using VideoEval for evaluation, it lacks comprehensive code for the entire processing flow (such as the processing flow mentioned in 3.1)
- Table 3 provides a comparison of various fine-tuning methods to reduce overhead, but they all belong to the category of low-parameter fine-tuning. It is possible to add some other routes of efficient fine-tuning methods such as MeZO[1] and HiZOO[2].
- The observation that “more video data can sometimes negatively affect certain tasks” is intriguing (Finding 2), however, there is no further analysis in the paper.

[1] Sadhika Malladi, Tianyu Gao, Eshaan Nichani, Alex Damian, Jason D. Lee, Danqi Chen, and Sanjeev Arora. Fine-tuning language models with just forward passes. In Thirty-seventh Conference on Neural Information Processing Systems, 2023.

[2] Yanjun Zhao, Sizhe Dang, Haishan Ye, Guang Dai, Yi Qian, and Ivor W. Tsang. Second-order fine-tuning without pain for llms:a hessian informed zeroth-order optimizer. In International Conference on Learning Representations, 2025.

**Strengths Contributions:**

- This paper is well-written, the tables are well-designed and contain a lot of information. Figure 2 clearly illustrates the process of building the dataset.
- By covering eight adaptation tasks and four embedding tasks drawn from varied domains (science, content moderation, aesthetics, emotion, etc.), VideoEval goes well beyond action‐recognition-only benchmarks like Kinetics or Something-Something.
- The training-light VidTAB and training-free VidEB protocols require orders of magnitude fewer samples than end-to-end retraining. This makes large-model evaluation feasible even for research groups without extensive compute.
- Unlike prior work focusing on single tasks or action recognition alone, VideoEval systematically evaluates both adaptability (VidTAB) and embedding utility (VidEB). And the large-scale study of 20 models yields actionable insights.

---

> ### Author Rebuttal · Authors · 2025-07-29
>
> **Q1. Although VidTAB spans eight tasks, it omits spatio-temporal localization and generative video tasks.**
>
> Regarding spatio-temporal localization, due to the significant discrepancy between this task and the tasks involved in the pre-training of VFMs, we find it challenging to perform few-shot evaluation. Specifically, a small amount of data is insufficient to effectively train a detection head.
>
> For generative video tasks, the requirement for additional text decoders, and in such cases, the evaluation transforms into multi-modal evaluation. First, we would like to highlight the key differences between single-modal (vision-centric) and multi-modal evaluations, taking interaction with the language modality as an example:
>
> 1. For single-modal (vision-centric) evaluation, since the task heads used by all VFMs are identical and very lightweight, the evaluation results is **solely determined by the VFMs themselves**.
> 2. For multi-modal evaluation, however, the involvement of language models complicates the assessment of VFMs, with two main approaches:
> 	- Using language models that are originally matched with the VFMs to be evaluated (e.g., using CLIP-Text, which corresponds to CLIP-ViT). A notable issue here is **unfair comparison**, as different language models can obviously affect the final evaluation results. For instance, in LiT[1], significant improvements can still be achieved by freezing the ViT while only training the text encoder.
> 	- The other approach is adopting the same language model, i.e., integrating different VFMs with the same language model. For example, integrating with large language models to assess their performance in generative tasks such as question answering. This approach has **two major drawbacks**: (1) Excessively high training costs, as sufficient data and training time are required to converge to reasonable results; (2) Introducing overly powerful language models may bias the evaluation of VFMs. For example, consider VFMs A and B: although A has stronger temporal modeling capabilities than B, when performing multi-modal generative tasks like video captioning, using an LLM with strong temporal modeling as the text decoder may mask A’s advantages over B, thereby biasing the assessment of the VFMs’ inherent capabilities.
>
> In summary, given that our goal is centered on evaluating the VFMs themselves, our VideoEval focuses exclusively on single-modal evaluation. Furthermore, for such generative tasks, the community already has a multitude of benchmarks (e.g., MVBench[2], VideoMME[3]). However, there is a lack of benchmarks like our VideoEval, which directly conduct vision-centric evaluations on VFMs. We believe it is complementary to multi-modal evaluation, and a more comprehensive assessment can be achieved by combining it with existing multi-modal evaluation methods.
>
> **Q2. Although the paper provides detailed code for using VideoEval for evaluation, it lacks comprehensive code for the entire processing flow.**
>
> Thank you for your valuable suggestions. We have supplemented the complete processing workflow in the code repository; see "How to build VideoEval".
>
> **Q3. It is possible to add some other routes of efficient fine-tuning methods such as MeZO[1] and HiZOO[2].**
>
> Thank you for your valuable suggestions. In our paper, we evaluated only those fine-tuning strategies that have been fully validated by previous studies as applicable to Visual Foundation Models. Consequently, we did not consider the fine-tuning strategies you mentioned, which may be widely used exclusively in Language Models. We will endeavor to apply them to the adaptation of VFMs in future work.
>
> **Q4. The observation that “more video data can sometimes negatively affect certain tasks” is intriguing (Finding 2), however, there is no further analysis in the paper.**
>
> We apologize that the analysis in the paper is not sufficiently detailed due to space constraints, and we provide supplements here: First, as stated in the paper, we attribute the cause of this phenomenon to the insufficient diversity of training data, that is, most commonly used video data are predominantly human- and action-centric. One of the most direct pieces of evidence is as follows: ViCLIP-L is initialized based on CLIP-L and undergoes post-training on the large-scale video-text pair dataset InternVid200M[4]. Nonetheless, in our evaluation, only its Action-related scores surpass those of the original CLIP, while scores on other tasks such as Science, Safety, Quality, and Emotion show no improvement or even a significant decline. However, in the evaluation of its original paper, traditional benchmarks such as Kinetics all demonstrate its significant improvement over CLIP. We believe this indicates that our VidTAB can evaluate capabilities that were not assessed by previous benchmarks and can, to some extent, guide the curation strategies in the construction of large-scale pre-training datasets like InternVid.
>
> | Model  | Pretrained Weight/Data       | Kinetics400-FT | SthSthv2-FT | VidTAB-Action         | VidTAB-Science        | VidTAB-Safety         |
> | ------ | ---------------------------- | -------------- | ----------- | --------------------- | --------------------- | --------------------- |
> | CLIP   | from scratch / CLIP400M      | 86.7           | 70.1        | 31.9/37.8             | 32.3/37.4             | 54.2/58.2             |
> | ViCLIP | CLIP / InternVid10M-Filtered | 86.8(+0.1)     | 71.2(+1.1)  | 31.2(-0.7)/42.7(+4.9) | 30.2(-2.1)/35.3(-2.1) | 47.9(-6.3)/53.9(-4.3) |
> |        | CLIP / InternVid200M         | 87.9(+1.2)     | 73.6(+3.5)  | 38.2(+6.3)/44.7(+6.9) | 30.2(-2.1)/37.9(+0.5) | 47.4(-6.8)/54.9(-3.3) |
>
> [1] LiT: Zero-Shot Transfer with Locked-image text Tuning
>
> [2] Mvbench: A comprehensive multi-modal video understanding benchmark
>
> [3] Video-MME: The first-ever comprehensive evaluation benchmark of multi-modal llms in video analysis
>
> [4] InternVid: A Large-scale Video-Text Dataset for Multimodal Understanding and Generation

---

> > ### Comment · Reviewer_e8Pz · 2025-08-05
> > **Thanks for your detail response**
> >
> > The author's responses address most of my concerns, and I believe these changes will contribute to the final quality of the paper. I will maintain my initial score.

---

> > > ### Author Response · Authors · 2025-08-06
> > >
> > > Thank you for your reply and valuable suggestions. We will incorporate the additional analysis and explanations you asked us to provide, and will conduct a thorough re-polishing of the writing structure and content of the paper.

---

> ### Comment · Area_Chair_7w7c · 2025-08-04
>
> Dear e8Pz,
>
> Please make sure to read the other reviews and the author response and engage in an open exchange with the authors asap so there is time for back and forth discussion.
>
> Best, AC

---

### Official Review · Reviewer_VKcc · 2025-07-03

**Rating:** 5
**Confidence:** 3

**Summary:**

This paper presents a comprehensive benchmark suite for vision foundation models (including image embedding models, video embedding models, and vision-language models). The benchmark consists of two parts: (1) A set of few-shot classification tasks, where model performance is reported as an average over different numbers of fine-tuning/adaptation shots, and (2) a hierarchical retrieval task that measures whether model embeddings accurately reflect video similarity from direct copies, to different shots of the same scene. The authors evaluate a host of open, state-of-the art vision foundation models, finding that models might still exhibit weak generalization, and do not necessarily benefit from more training data.

**Additional Feedback:**

- L36 Previous -> previous
- L39 evaluates -> evaluate
- Model labels and numbers in Fig. 1 are too small. The white font on the videos and black font on the eye have low contrast and are hard to read.
- Fig. 1: why are the scales different on each axis of the radar plot (e.g., Kinetics seems to  show ~90 on the outer circle, while VidTAB-Dark shows ~50)? This makes it very hard to compare results and assess saturation at a glance, please fix. Ideally, each grid line should correspond to a round value like 0-25-50-75-100.
- Fig. 1: I would advise to use a clean, simple, and straight font appropriate for a scientific paper.
- L69 _are_ as follows...
- Tab. 2 fontsize is too small
- Fig. 2 and Tab. 2 grey font has low contrast and is hard to read
- Fig. 3 is also very small. Please increase to page width.
- It would be helpful to split the # Pt. Data column up into 2~3 colums: I, V, "Kinetics used" (check/cross) for easier parsing
- The result sections are very hard to parse. It would be beneficial to break them up with paragraphs, or as an enumeration, and adapt the formatting (e.g., make the main takeaway bold (which is now in italics)

**Dataset Code Accessibility:**

Yes

**Dataset Code Comments:**

I checked the dataset and github.

**Ethical Considerations:**

No, there are no or only very minor ethics concerns

**Final Justification:**

The benchmark is comprehensive and well-suited for evaluating foundation models in a few-shot setting; the results will be of interest to the community.

**Limitations Weaknesses:**

The paper is mostly held back by the presentation of the results and some missing details. I hope these shortcomings can be addressed during the rebuttal.

1. It is not entirely clear to me why benchmarks like InternVideo, Kinetics, or VideoGlue might not also be adapted to have lower validation cost. E.g., one could measure zero- or few-shot performance on these benchmarks as well or do zero-shot clustering instead of classification, avoiding most of the training cost. I'd be curious to hear the authors' thoughts on this.
2. It would be nice to also include audiovisual classification datasets like AudioSet and VGGSounder in the related works, which can also be used for benchmarking and are orthogonal to, e.g., action understanding.
3. Since the benchmark focuses on tasks that are discriminate *now*, I worry that it might not be very future proof. It would be nice to have various difficulties for each task, with performance on the highest difficulty close to random chance. An easy "knob" to control difficulty could be the number of categories, as hinted at in LL139.
4. It is unclear to me why all tasks are converted to a classification task. Wouldn't it be just as simple to have a lightweight regression head or report multi-label classification metrics where necessary? Additionally, the number of classes is generally low, so that I am worried some results might be rather noisy.
5. Since the number of classes varies across tasks, random chance performance should be reported. Even better, performance could be reported relative to random chance, to facilitate comparison across tasks.
6. I am curious what a plot of fine-tuning/adaptation compute (e.g., number of parameters × number of update steps) vs. performance would look like compared to the plot of performance vs. number of shots (Fig. 3).
7. Tab. 3 is somewhat unclear. Am I correct in assuming that the numbers here correspond to zero-shot performance as indicated in Sec. 4.2?
8. I understand that evaluation of proprietary models is costly and impossible in most cases where adaptation is required. It might be interesting, however, to include zero-shot performance of some proprietary models (e.g., Veo) in Tab. 3, if only for a subset of the data points (e.g., 10%).
9. For main result (2) it would be beneficial to show plots (performance vs. model size / data points) to more easily assess this point.

**Strengths Contributions:**

1. The benchmark is comprehensive and well-suited for evaluating foundation models in a few-shot setting.
2. The comparisons are thorough and include a large number of relevant models
3. The findings are novel and interesting, and, I believe, will be of interest to the community.

---

> ### Author Rebuttal · Authors · 2025-07-29
>
> **Q1. Why benchmarks like InternVideo, Kinetics, or VideoGlue might not also be adapted to have lower validation cost?**
>
> We concur with your suggestion that prior benchmarks could be evaluated under a similarly low-cost setting. Nevertheless, two issues remain. First, datasets such as Kinetics are comparatively homogeneous: they are overwhelmingly centered on human action recognition. Second, because these datasets possess large-scale training splits, models including VideoMAE and InternVideo inevitably incorporate them during pre-training, precluding evaluation of out-of-domain generalization. VideoEval is therefore intended as a complementary benchmark that explicitly probes model performance on out-of-domain videos.
>
> **Q2. It would be nice to also include audiovisual classification datasets in the related works.**
>
> Thank you for your suggestion, we will include audiovisual classification datasets in the related works. We agree that comprehensive video understanding necessitates the inclusion of audio. However, current video foundation models do not natively integrate audio; instead, they rely on separate encoders for audio tasks, which precludes joint evaluation. We will therefore pursue the development and evaluation of audio-visual foundation models in future work.
>
> **Q3. It would be nice to have various difficulties for each task, with performance on the highest difficulty close to random chance.**
>
> Thank you for your suggestion. Although we calibrated task difficulty to maintain discriminative power among existing models, most tasks remain extremely challenging: even the strongest closed-source model to date—Gemini 2.5 Pro, whose results we reported in response Q8—achieves only poor performance. We believe the current benchmark is already sufficiently demanding. Nonetheless, we fully agree that benchmarks must evolve alongside increasingly capable models, and we will actively develop more challenging tasks in future work.
>
> **Q4. It is unclear to me why all tasks are converted to a classification task.**
>
> We reformulated all tasks as classification problems for two reasons. First, doing so mitigates the subjective bias inherent in certain annotations: while raters might disagree over whether a video merits a quality score of 9 versus 8, almost unanimous consensus exists when comparing a score-9 video with one rated 1. Second, the reformulation streamlines evaluation metrics, following VTAB[1]. Moreover, the deliberately small number of classes serves to calibrate task difficulty. By preventing the tasks from becoming prohibitively hard, we avoid the degenerate scenario in which all models perform at random, thereby ensuring meaningful results.
>
> **Q5. Random chance performance should be reported.**
>
> We appreciate your insightful suggestion. Random performance has already been reported in Table 5, and we will further refine the tabular presentation by adopting relative improvement as an additional metric.
>
> **Q6. A plot of fine-tuning/adaptation compute vs. performance**
>
> Thank you for your suggestion. Due to rebuttal-format constraints we are unable to include the requested plots here; the complete figures will be provided in the final version. In the meantime, we supply the 16-shot performance curve of V-JEPA-H on Action Recogntion in Dark against training update steps for your reference.
>
> | Adaptation method | Tunable Params (M) | 5 epoch | 10 epoch | 15  epoch | 20  epoch | 25 epoch |
> | ----------------- | ------------------ | ------- | -------- | --------- | --------- | -------- |
> | linear probe      | 0.0                | 7.2     | 7.8      | 8.4       | 9.4       | 9.6      |
> | attentive probe   | 19.7               | 32.5    | 44.8     | 47.4      | 51.5      | 52.9     |
> | adapter           | 52.6               | 14.5    | 35.6     | 62.4      | 66.0      | 71.8     |
> | full finetuning   | 663.7              | 20.1    | 59.2     | 78.1      | 80.6      | 79.9     |
>
> We observe that adaptation performance exhibits a significant positive correlation with fine-tuning cost (in terms of the number of parameters and training steps).
>
> **Q7. Does Tab. 3 report zero-shot performance (Sec. 4.2)?**
>
> Yes, the results in Table 3 were obtained using the same zero-shot evaluation protocol described in Section 4.2.
>
> **Q8. zero-shot performance of some proprietary models.**
>
> Thank you for the suggestion. Although our focus is vision-centric, we recognize that proprietary models can still be leveraged to gauge zero-shot performance. We therefore sample approximately 50 examples per task (balanced across classes) and evaluate them with both Gemini 2.5 Flash and Gemini 2.5 Pro[2], we additionally include human evaluations for comparison. The results are reported as follows:
>
> | Model           | Date  | AR in Dark Scene | AR in Long Video | Medical Surgery | Animal Behavior | Fake Face | Harmfull Content | Quality Assess | Emotion Analysis |
> | --------------- | ----- | ---------------- | ---------------- | --------------- | --------------- | --------- | ---------------- | -------------- | ---------------- |
> | random          | -     | 9.1              | 10.0             | 6.3             | 8.3             | 50.0      | 33.3             | 50.0           | 14.3             |
> | human           | -     | 75.0             | 85.0             | 58.3            | 70.8            | 87.5      | 66.7             | 81.3           | 53.6             |
> | InternVideo2-1B | 24.03 | 37.1             | 40.2             | 11.5            | 45.2            | 51.3      | 59.1             | 56.1           | 24.3             |
> | Gemini2.5-Flash | 25.06 | 34.1             | 70.0             | 22.9            | 52.1            | 62.5      | 41.7             | 71.9           | 42.9             |
> | Gemini2.5-Pro   | 25.06 | 43.2             | 70.0             | 28.1            | 56.3            | 65.6      | 50.0             | 59.4           | 46.4             |
> > Note that for human evaluation, we only asked the evaluators to judge the videos based on category names, so the accuracy may be reduced due to their lack of relevant background knowledge. Regarding the reliability of the original annotations, since the annotation information we used is either constructed from real information or manually annotated, and we have conducted manual sampling checks on these annotations during task selection, the reliability of their quality is ensured.
>
> As shown in the table above, even the current state-of-the-art video-understanding models still fall far short of solving our tasks. Even for a human who is unaware of the task background and lacks some of the knowledge required for the task, our task remains highly challenging.
>
> **Q9. Regarding issues of writing and presentation.**
>
> We thank you for the constructive feedback and will refine the manuscript accordingly in the next revision.
>
> [1] A Large-scale Study of Representation Learning with the Visual Task Adaptation Benchmark
>
> [2] Gemini 2.5: Pushing the frontier with advanced reasoning, multimodality, long context, and next generation agentic capabilities

---

> ### Comment · Area_Chair_7w7c · 2025-08-04
>
> Dear VKcc,
>
> Please make sure to read the other reviews and the author response and engage in an open exchange with the authors.
>
> Best, AC

---

> ### Author Response · Authors · 2025-08-09
>
> Dear Reviewer VKcc,
>
> Thank you for your thoughtful feedback and supportive comments on our work!
>
> We have made effort to address your concerns in our previous response. As the author-reviewer discussion period is about to end, we would greatly appreciate it if you could let us know whether our reply sufficiently resolves your questions. If you have any further comments or concerns, we would be more than happy to discuss them.

---

### Decision · Program_Chairs · 2025-09-18

**Decision:**

Reject

**Comment:**

This paper proposes VideoEval to comprehensively evaluate video foundation models which evaluates the the task adaptability of video foundation models under few-shot and zero-shot settings. Initially this paper received diverging ratings (2 accept, 1 borderline accept, 1 strong reject). Reviewers appreciated the comprehensive nature of the benchmark (VKcc, e8Pz, 6mG5 ), the large number of models compared (VKcc, 6mG5), the findings that resulted from this benchmarking (VKcc, e8Pz) and low compute nature of the benchmark (e8Pz, 6mG5). However, reviewers had several concerns including the potential to future proof the dataset with adaptable difficulty (VKcc), missing random chance baseline (VKcc), lack of certain types of datasets including multimodal tasks (VKcc, e8Pz, 6mG5), lack of other efficient fine-tuning methods (e8Pz), lack of human evaluation (Hwmv).

After the rebuttal the reviewers felt most of their concerns were addressed, with the only remaining concerns relating to the writing quality. This led to final scores of (2 accept, 1 borderline accept, 1 borderline reject). The AC agrees with the positive assessment of the reviewers and that this paper should be accepted. The authors are strongly encouraged to address the clarity issues and fix the typos highlighted by reviewers in the final version.

===== FINAL UPDATE FROM DB Track PCs ====

The final decision for this paper has been taken by the program chairs after consultation with the SACs. All Senior Area Chairs have ranked papers according to the feedback from the AC during the review process. We decided to leave the original meta-review to reflect the opinion of the AC in light of the initial discussions with reviewers and SAC.